# Trait and Ability Emotional Intelligence and Its Impact on Sports Performance of Athletes

**DOI:** 10.3390/sports9050060

**Published:** 2021-05-10

**Authors:** Alexandra Kopp, Markus Reichert, Darko Jekauc

**Affiliations:** 1Department of Sport Science, Humboldt University of Berlin, D-10115 Berlin, Germany; alexandra.kopp@hu-berlin.de; 2Department of eHealth and Sports Analytics, Ruhr University Bochum, D-44801 Bochum, Germany; markus.reichert@partner.kit.edu; 3Department of Sport Science, Karlsruhe Institute of Technology, D-76131 Karlsruhe, Germany

**Keywords:** emotional intelligence, trait EI, ability EI, MSCEIT, TEIQue, sports performance

## Abstract

Emotional intelligence (EI) is considered a determinant of sports performance. Two opposing perspectives have been discussed in the theoretical discourse on EI: EI as an ability versus EI as a trait, both widely differing in content and method of assessment. Previous applied sport psychology research is characterized by a heterogeneous use of different conceptualizations and measurements of EI. However, evidence for the superiority of an EI concept does not exist. This study directly compares the ability and trait EI concepts in the relationship with athletic performance. An online survey was conducted (response rate = 19%). Participants completed the Mayer-Salovey-Caruso Emotional Intelligence Test, the Trait Emotional Intelligence Questionnaire Short Form, a list of questions about biographical information as well as information related to sports performance and sport participation. We used regression analyses and controlled type of sports to investigate how sports performance is influenced by EI. Trait EI positively predicted self-assessment of athletes’ performance (B = 1.02; *p* < 0.01) whereby ability EI did not predict any outcome of sports performance. The effect of trait EI was independent of the ability EI. Overall, the result indicates some evidence for the superiority of the trait EI in applied sports psychology.

## 1. Introduction

Emotions are an inextricable feature of human experience, behavior and interaction and, therefore, a fundamental part of our human nature [1,2]. Consequently, it is hardly surprising that emotions are also an inherent component of the competitive experience in sports [3,4,5]. A wide range of studies demonstrates that emotions can either facilitate or deteriorate sports performance, depending on their content, time occurrence and intensity level [3,5,6]. Therefore, emotions have an important impact on athletic performance [7]. In addition, the capacity of athletes to perceive emotions, understand and manage them, in an effort to optimize their sports performance, appears to be an essential part of athletic sports success [1,6,8,9]. In accordance with this assumption, there is a growing interest in applied sports psychology research to analyze the potential influence of EI on sports performance.

EI is defined as “the ability to perceive and express emotion, assimilate emotion in thought, understand and reason with emotion and regulate emotion in the self and others” [10] (p. 396). In the EI research history, the analysis of individual variations of how individuals recognize, interpret, express, manage and utilize their own emotions and those of others is commonly the focus [11]. EI is regarded “as a distinct construct from traditional IQ and personality, which facilitates the potential for prediction of and influence on, various real-life outcomes” [12] (p. 25). Since 1990, research on EI has drastically grown and EI has become a hot topic both inside and outside scientific literature [13,14,15,16,17]. However, despite its high public profile, EI has also been constantly accompanied by considerable scientific criticism with regard to its concept, theory and measurement [18,19,20].

The field of EI research split off into two distinct perspectives concerning the conceptualization of EI: the trait and the ability approach [17]. EI was originally conceived as a cognitive capability, relying on the processing of affective content and includes a number of skills that can be trained and enhanced through time. The ability EI perspective conceives EI as “the cooperative combination of intelligence and emotion” and the four branch model, implemented by Mayer and Salovey [11], is viewed as the general model of ability EI [12]. Ability EI is evaluated with IQ-like performance tests. Based on a number of hypothetical scenarios that have to be solved, the ability EI of the individuals is measured. The Mayer–Salovey–Caruso Emotional Intelligence Test (MSCEIT) is the most commonly applied ability-measurement and has been the only test available for a long time [12,21].

The trait theory was introduced by Petrides [22]. In this, EI “represents a constellation of emotional perceptions located at the lower levels of personality hierarchies” [23] (p. 261) and is viewed as a kind of disposition, which reflects the way people ordinarily act in emotional situations [17]. In this sense, the trait theory implies that it is not about acting in the right way or solving emotional scenarios. Instead, it is about how individuals believe they can cope with emotional situations [24]. This is typically assessed through questions and rating scales, for instance with the Trait Emotional Intelligence Questionnaire (TEIQue) [24]. For completeness, it should be mentioned that research endeavors exist, integrating both perspectives. For instance, the Tripartide model [17] and the multi-level investment model of EI [25]. However, these models were not the focus point of the current study and therefore we do not provide a detailed description.

Competitive sport is characterized by the fact that athletes are constantly faced with emotional demands that can have an influence on their sports performance. The ability to perceive, understand and reason one’s own emotions and those of teammates, trainers, referees, fans and competitors is therefore necessary to achieve optimal athletic performance [26,27]. Previous research examined the impact of EI on team and individual performance indicators in particular sports (e.g., canoeing, cricket, ballet, baseball, basketball, hockey, ice hockey, soccer, tennis, volleyball) and the results are quite different. For instance, regarding the subjective level, Laborde et al. [28] showed that among athletes from both individual and team sports, higher performance satisfaction is associated with a higher EI. Likewise, Petrides et al. [29] found a positive relationship between EI and ballet dancing ability ratings. Considering objective performance parameters in sport, Crombie et al. [30] demonstrated that team ability EI of cricketers was positively related to the team performance measured by game statistics. In baseball, the EI was also moderately associated with pitching performance, but not with batting performance throughout the whole season [31]. On the other hand, Perlini et al. [32] found no relation between positive game statistics and EI in ice hockey athletes. Furthermore, regarding the expertise level as a performance parameter in sport, Saies et al. [33] observed a positive correlation between EI and expertise level classified by sports results in national championships, while Laborde et al. [28] found no significant effects with level of expertise assessed by self-report.

Findings of the meta-analysis on the influence of EI on sports performance demonstrated a small positive association [34]. The authors concluded that EI can currently be considered as a weak determinant of performance in sports; however, the authors also indicated that more exploitable studies are required to allow for better understanding.

To date, a diverse use of varying conceptualizations and methods of measuring EI and also an almost exclusive use of self-report questionnaires typify the applied sport psychology research [7,34]. This was clearly illustrated by Kopp’s and Jekauc’s [34] meta-analysis. They identified 14 (out of a total of 17) different measurements of EI. Only two studies used an ability-test for measuring EI, while in all other studies self-report tests were utilized [30,35]. So far, no agreement has been reached on the definition, model and measurement of EI. For example, Meyer and Fletcher [36] suggest in their theoretical overview to consider the ability model for future applied sport psychology research. In contrast, Laborde et al. [37] recommend the trait approach. Finally, meta-analytic results do not indicate a preference for either the ability or the trait EI concept [34]. Until now, there is no study to comparing the effectivity and usefulness of EI measures to examine the existing association between EI and sports performance.

Therefore, the goal of our investigation was to analyze the relationship of ability and trait EI with sports performance outcomes in competitive sports. The reason for conducting this analysis was to provide the first direct comparison of ability and trait EI concepts in the relationship between EI and sports performance. To our best knowledge, no prior analyses assessed EI by both self-report and performance measure of EI in an applied sport psychology context. The current study aims to address this research gap. Specifically, we expect a positive effect of trait EIand a positive effect of ability EI on sports performance. Additionally, we explored the relative effect of ability and trait EI on performance in sports.

## 2. Materials and Methods

### 2.1. Participants

The study included a total of 323 (52.9% male) athletes who were actually active in competitive sports at the time of the survey. Individuals who were not but, participated in the survey were not included in the study. Almost all athletes were German nationals. With regards to age, 15.2% were 11–20 years old, 48.3% were 21–34 years old and 36% were older than 34 (*M*_age_ = 33.4 years; *SD* = 14.2). They had been practicing their sports for an average of 16.4 years (*SD* = 12.0) and were training on average 8.9 h a week (*SD* = 6.7), with an average training session of 4.2 h a week (*SD* = 2.8). The number of athletes who regularly participated in competitions was 247 (76.5%). Finally, a total of 56 different sports were represented in this study.

### 2.2. Procedure

An online survey was conducted with the survey period lasting from 1 February 2019 to 27 June 2019. A total of 44 federal professional associations (Bundesfachverbände), 132 regional associations (Landesfachverbände), 16 national federations of sports (Landessportbünde) and 121 sport clubs were contacted by e-mail (and/or telephone) and asked to support our study. Coaches and athletes were also contacted directly and asked to support or to participate respectively in our survey. Finally, contacts in sports institutes as well as the social media platform Facebook were also used for recruitment. Email content included information about the current study and also a pre-formulated call for participation in the study, which was to be forwarded on to the members. The processing of the questions took about 30 min. Participation in this study was voluntary and everyone had the right to withdraw at any time. Data were anonymized. Participants completed the measures in the following order: demographics, MSCEIT and TEIQue. In total, the link to the survey was clicked 1659 times, 323 athletes completed the survey (response rate of 19.5%) and 717 athletes did not finish the survey.

### 2.3. Instruments

#### 2.3.1. The Mayer–Salovey–Caruso Emotional Intelligence Test (MSCEIT)

To evaluate ability EI with performance tests, we used the Mayer–Salovey–Caruso Emotional Intelligence Test (MSCEIT) [38] in its German version [39]. The MSCEIT is the most used ability EI test in psychology research and it is currently, considered to be the most appropriate ability test in an applied sport psychology context [7]. The MSCEIT measures across the four branches of EI how effectively individuals handle emotionally charged situations. The MSCEIT comprises a total of 141 items which form a total score and four dimension scores: (a) perceiving emotions PE, (b) using emotions to facilitate thought FT, (c) understanding emotions UE and (d) managing emotions ME. The MSCEIT is scored with both consensus and expert scoring methods. Correlations between expert and consensus methods for the German version of the MSCEIT™ are high (0.89 to 0.99), although slightly lower than the American comparison. Reliabilities (Cronbach’s *α*) and split half reliabilities were 0.90/0.93 for PE, 0.66/0.77 for FT, 0.73/0.72 for UE, 0.68/0.73 for ME [39].

#### 2.3.2. Trait Emotional Intelligence Questionnaire Short Form (TEIQue-SF)

The TEIQue-SF (German version) was utilized in this study, to evaluate the trait EI [40], whose factor structure has been verified for sports [28]. This inventory contains 30 items and four factors: well-being, self-control, emotionality and sociability [24]. The TEIQue-SF aims to assess the individual’s self-perceived abilities and behavioral dispositions on a 7-point Likert scale format ranging from 1 (completely disagree) to 7 (completely agree) [40]. In the current study, Cronbach’s *α* reliability coefficients for the TEIQue-SF were: 0.77 for well-being, 0.62 for self-control, 0.63 for emotionality, 0.60 for sociability and 0.84 for the total score, see Appendix A.

#### 2.3.3. Sports Performance

To evaluate sports performance across different types of sports, we used three indicators, all measured by self-report. As it was difficult to agree on a common expertise indicator for all type of sports, we applied this method. First, we assessed expertise level via self-report by asking the participants to indicate in which league they participate in competitions. (e.g., (1) international-level, (2) national-level, (3) regional-level, (4) district-level, (5) and underlying-level). However, the categorization into five levels turned out to be too weak. Therefore, for statistical analyses we coded into two categories: (1) international competitive level (elite athletes who were competing at both national and international level) and (2) national competitive level (athletes solely competing on a national level). This procedure assumes that the athletes competing at an international level are the elite in their country in their respective discipline. Although it is not free of criticism, it is appropriate within the sports field and seems to be a valid way of defining elite athletes [41].

Second, we assessed sports success by self-report. In this item, participants reported their greatest success in their career. Again, we coded into two categories for statistical analyses: (1) international sports success and (2) national sports success. As well as level of expertise in sports, the highest sport success of the athletes is also a useful reference for their expertise and, ultimately, their sports performance [41].

Third, sports performance was measured via a self-assessment of athletics performance. This variable allows to compare the achievement of athletes competing in several sports with very different level of expertise and acting in different positions [28,42]. The four items (e.g., “How performance-oriented are you in sports?”; “How do you rate your athletic performance?”; “The satisfaction with my athletic performance is ...”; “The level of my athletic performance is currently ...”) are completed on a 5-points Likert scale ranging (1 (low) to 5 (very high)). Reliability in our investigation was 0.76.

### 2.4. Data Preprocessing

First, for analyses on expertise level and sport success, we built two variables for each performance parameter. We used the information reported by participants (see Section 2.3.3 and parameterized a dichotomous variable for indicating level of expertise as well as another dichotomous variable for indicating the greatest sport success. These variables were coded “0” if participants had reported that they competed on an international championship (level of expertise) or, if participants reported they had international sports success (success at important international championships, such as the World Championships, European Championship or Olympic Games (e.g., records, titles or medals)). The variables were coded “1” if this was not the case (competing or success at national or underlying level). These two variables and the continuous variable, self-assessment of athletic performance, entered our regression analyses as outcomes of interest. Because athletes interact in their “type of sport” and, therefore, athletes are influenced by the type of sport to which they belong to as well as the properties of those groups are in turn influenced by the athletes who make up that group [43], we classified the type of sports into three groups: individual sports athletes without a direct opponent, individual sports athletes with a direct opponent and team sports athletes, following the categorization of Laborde [44].

### 2.5. Statistical Data Analysis

Meta-analytic evidence of the EI–sports-performance relationship suggests a significant positive correlation (*R* = 0.19) of the relationship between ability approach to EI and also trait approach to EI and sports performance [34] (p. 15). A priori power analysis with G*Power for a linear multiple regression (fixed model, *R*^2^ deviation from zero) with an estimated effect size (*R* = 0.19; *f*^2^ = 0.037), a power of 0.80, an alpha level of 0.05 and two predictors estimated a necessary sample size of *n* = 261 [45]. We checked all assumptions for linear multiple regression. The linear relationship between the independent and dependent variable was examined by scatter plots, which indicated non-linearity. All numerical variables and residuals were checked for normality using the Kolmogorov–Smirnov test, the Shapiro–Wilk test and Q–Q plots. Results indicated that the data were not normally distributed. The residual scatter plots presented a picture of homoscedasticity and a white test confirmed this graphical estimation. Independent variables demonstrated no or partly little multicollinearity. The Durbin-Watson test was performed to check auto-correlation. Variables were not auto-correlated. Finally, due to assumption violations, analyses with non-parametric statistics were run. To analyze whether ability EI and trait EI predicted sports performance and to assess if effects of both predictors vary, we carried out linear as well as logistic regression analyses. We entered ability EI and trait EI as predictors and sports performance as outcome measure, operationalized with three different indicators (see Section 2.4). Logistic regressions were conducted to estimate the effect of ability and trait EI on level of expertise (national vs. international level) and sport success (national vs. international sport success) [46]. Linear regressions were performed to assess the effect on ability and trait EI on self-assessment of athletic performance. In all regression models, we used dummy variables for the three groups (individual sports athletes without a direct opponent, individual sports athletes with a direct opponent and team sports athletes) to control for the type of sport. Results of the logistic regression are presented as adjusted odds ratios (*OR*), wald statistic (*χ*^2^), *p*-values and the Nagelkerke’s *R*^2^ (Pseudo-*R*^2^ values) and linear regression analyses are presented as the coefficient of regression, *t*-values, *p*-values and the percent of explained variance (*R*^2^). The alpha level was set to *p* < 0.05. All statistical analyses were performed in IBM SPSS Statistics for Windows, Version 25.0 (Armonk, NY, USA: IBM Corp).

## 3. Results

### 3.1. Descriptive Results

As expected, the majority of the participants (67%) competed in a national championship (national expertise level), while 10.2% of the athletes in our sample had an international expertise level (for 22,3% it was not possible to make an allocation). The same picture emerges regarding the sports success of the participants. Eighty percent achieved success on a national level and 20.1% reported sports success on an international level. With regards to the self-assessment of athletic performance, the sample achieves an average value of 14.2 (*SD* = 2.96) (possible ranging from 4 to 20). Thus, the score of our sample lies in the average range in relation to the norm group of this inventory [39,40]. First, analyses regarding the EI resulted in a mean value of 5.21 (*SD* = 0.58) for the trait EI and a mean value of 99.85 (*SD* = 15.97) for the ability EI. The overall descriptive statistics on sociodemographic information, EI levels and performance levels also related to the different types of sports are depicted in Table 1.

### 3.2. Association of Trait Emotional Intelligence on Sports Performance

To test whether trait EI does predict athletic sports performance, we calculated one regression model for each sport performance outcome measure, resulting in a total of three models (two logistic and one linear regression model, see Section 2.5). As hypothesized, trait EI positively predicted self-assessment of athletes’ performances (*B* = 1.02; *t*(1) = 3.69; *p* < 0.01; *R*^2^ = 0.041). Put into practice, an increase on the trait EI scale by 1 point (on a scale ranging from 1 to 7) was associated with an increase by 1.02 points of the evaluation of subjective performance (on a scale ranging from 1 to 5), indicating a considerable effect. However, trait EI did not significantly predict level of expertise (*OR* = 1.07; *χ*^2^(1) = 0.05; *p* = 0.83) and sport success (*OR* = 1.53; *χ*^2^(1) = 3.00; *p* = 0.08). The full regression matrix is presented in Table 2 and Table 3.

### 3.3. Association of Ability Emotional Intelligence on Sports Performance

To test whether ability EI does predict athletic sports performance, we again calculated one regression model for each sports performance outcome measure, resulting in a total of three models (two logistic and one linear regression model, see Section 2.5). However, contrary to our assumptions (hypothesis II), ability EI did not predict level of expertise (*OR* = 1.01; *χ*^2^(1) = 0.71; *p* = 0.40), sport success (*OR* = 0.98; *χ*^2^(1) = 0.07; *p* = 0.80) nor self-assessment of athletics performance (*B* = 0.02; *t*(1) = 1.26; *p* = 0.21; *R*^2^ = 0.01; see Table 2 and Table 3).

### 3.4. Relative Association of Trait Emotional Intelligence and Ability Emotional Intelligence on Sports Performance

To explore the relative associations of ability and trait EI on performance, we entered both predictors into one regression model for each sport performance outcome measure, resulting in a total of three models (two logistic and one linear regression model, see Section 2.5). The overall regression model for self-assessment of athletic performance explains a variance of 4% (*F*(4218) = 2.16, *p* = 0.07). However, only trait EI significantly predicted self-assessment of athletic performance (*B* = 0.80; *t*(1) = 2.49; *p* = 0.01). Put into practice, an increase on the trait EI scale by 1 point (on a scale ranging from 1 to 7) was associated with an increase by 0.8 points of the evaluation of subjective performance (on a scale ranging from 1 to 5), indicating a considerable effect. Ability EI did not predict self-assessment of athletic performance (*B* = 0.01; *t*(1) = 1.23; *p* = 0.22). The level of expertise was not predicted by ability EI (*OR* = 1.01; *χ*^2^(1) = 0.59 *p* = 0.44), nor by the trait EI (*OR* = 1.26; *χ*^2^(1) = 0.45; *p* = 0.50) and the sport success of the athletes was not predicted by ability EI (*OR* = 1.00; *χ*^2^(1) = 0.18; *p = 0.67*), nor the trait EI (*OR* = 1.21; *χ*^2^(1) = 0.43; *p* = 0.51; see Table 2 and Table 3). Thus, the effect of trait EI on self-assessment of athletic performance was independent of the ability EI.

Finally, it should be noted, when controlling for different types of sports in accordance with the given classifications (individual athlete without direct opponent, individual athlete with direct opponent and team athlete), all results remained unchanged.

## 4. Discussion

This is the first investigation assessed with both self-report and performance measure of EI in an applied sport psychology context. The advantage of conducting a direct comparison of ability and trait EI concerning its relation to sports performance is that it can help provide a clear picture of the effectiveness and usefulness of each EI measurement. Results showed that trait EI positively predicted self-assessment of athletes’ performance (*B* = 1.02; *p* < 0.01;), whereby ability EI did not predict any outcome of sports performance. The effect of trait EI on self-assessment of athletic performance was independent of the ability EI.

First, regarding our results for trait EI, our assumption has been partially confirmed. In accordance with our prognosis, trait EI predicted the self-assessment of athletic performance. This result suggested that higher trait EI had been linked to higher self-assessment of athletic sports performance. Accordingly, a high trait EI score could have advantages for competitive athletes. This result encouraged the argument that trait EI can predict sports performance parameters like sports performance satisfaction, a variable representing a useful method of assessing the performance of athletes across various sports [42]. This is consistent with previous research using the TEIQue which demonstrated that athletes with upper trait EI scores also demonstrated better performance satisfaction [28]. Second, when controlling for different types of sports, results remained unchanged. This corresponds to the results of previous analyses [47,48] and shows that the trait EI is relevant for all three groups (individual athletes without direct opponent, individual athletes with direct opponent and team athletes). Furthermore, the explained variance for trait EI and self-assessment of athletic performance is small. Nevertheless, based on the results of the meta-analysis on the influence of EI on athletic performance, a small effect size can be expected [34]. However, the significant coefficients reflect the mean modification in the response for the predictor, while other predictors in the model are kept unchanged. Therefore, these kinds of information can be quite useful when viewed at the highest level of performance. Finally, it should be mentioned that, in this study, the self-reported behavior (self-assessment of athletic performance) was analyzed with the self-reported trait EI. Since it is the same type of measurement procedure, this may have influenced the effects and the results could be overestimated [49].

Furthermore, trait EI did not predict level of expertise and sport success. Because previous research had demonstrated that the regulation of emotions during competitions is important to achieve an effective performance [50], this result is not what we had anticipated. However, our results are not contradicting the mainstream findings [28,31,51,52,53], but contradict the conclusions of meta-analysis results [34], the systematic review [37] and relevant literature for this research area [44]. One explanation might be that trait EI is a component of various factors for generating of athletic performance and the influence of trait EI can vary depending on the requirement profile in each sport (1). Controlling for the three types of sports may still be too broad to correctly reflect the different emotional demands and finally identify possible effects. Additionally, trait EI sub-dimensions can lead to different effects on the relationship with sports performance and may even counter-balance each other. However, it is not shown when only the global trait EI is considered (2). Athletes in different competitive environments might need distinct trait EI characteristics and their exact association with athletic performance can differ depending on their sport and their environment [24]. Finally, it is in the nature of sports environments that there are always only a few athletes who participate in international competitions and show international success, which is also reflected in our data. These small group sizes may have had an impact on the results (3). However, our findings demonstrated that trait EI is not a special attribute of elite athletes and cannot predict the level of expertise and sport success.

Finally, there is another important aspect to discuss that relates to the question of whether our results are associated with personality traits. First, the fact that trait EI overlaps substantially with the higher order personality factors, and is therefore of little use, is a common criticism of the concept of EI as a personality trait [18,54,55]. Petrides et al. [56] argue that because “trait EI is explicitly conceptualized as a lower-order personality construct, it is expected to show strong correlations with the higher-order dimensions that define its factor space” (p. 28). So if a lower-order personality construct is not associated to the higher-order personality dimensions that define the factor space in which it lies, this would be rather unusual [56]. However, to address and verify this criticism, Andrei et al. [23] examined this question in a review and meta-analysis. The authors concluded that “although small, the overall effect size confirms the distinctiveness and theoretical importance of trait EI” (p. 272). They indicated that the TEIQue “consistently explains incremental variance in criteria pertaining to different areas of functioning, beyond higher order personality dimensions and other emotion-related variables” [23] (p. 261). Freudenthaler et al. [57] tested and validated the TEIQue in a German-speaking sample and their results supported the convergent and discriminatory validity of the TEIQue factors. Furthermore, the (criterion) validity was supported in the sports context. Laborde et al. [28] consider the TEIQue as a reliable instrument to assess EI in sports.

Regarding our results for ability EI, our hypothesis (2) was not supported by the evidence. The results of our study demonstrate that ability EI did not predict any of the sports performance outcomes. This indicates that ability EI is neither associated with an athlete’s expertise level nor with their sports success or self-assessment of sports performance. This finding is contrary to our expectations which are based on the link between emotions and performance, namely that emotions can either facilitate or debilitate sports performance [4,5,6,8,58]. Consequently, the ability to identify and understand emotions as well as regulating them are efficient methods in order to achieve optimal performance [5,59]. Although it is regarded as the most useful for the sports sector [36], there are only a few studies that have used the MSCEIT in the applied sport psychology context. Two studies could show positive associations between sports performance and ability EI [30,35]. In both studies, sports performance was operationalized as a team performance parameter. For this reason, a clear integration of our results into the current state of research is only conditionally applicable.

The points already discussed regarding trait EI (1–3), are also relevant for the ability EI. However, to avoid redundancy, we will not list them again.

Furthermore, there are certain aspects to discuss that relate to the ability of EI, measured by MSCEIT. First, the ability EI assessed by MSCEIT basically measures the knowledge dimension of EI and there can be some discrepancies between that and the actual day-to-day application of this knowledge in real social-emotional training and competition interaction. Accordingly, it is conceivable that an athlete with a high level of cognitive and verbal skills will be able to explain which emotional regulation strategies might be beneficial in a certain circumstance, although he/she may not be capable of applying the appropriate emotional regulation in a competitive or training situation [12]. Second, since the EI is a form of intelligence that relates to emotional spheres, Fiori et al. [12] suggest that ability EI predicted outcomes should be emotion-specific. They refer among other things to the meta-analysis of Joseph and Newman [60] which lead to the result that “ability EI positively predicts performance for high emotional labor jobs and negatively predicts performance for low emotional labor jobs” (p. 1). Following these findings, it would be more promising to use sport performance outcomes related to performance situations with a high emotional requirement profile and to examine their relationship to the ability EI. Finally, as it has been observed that the MSCEIT detects mainly individuals with lower EI level, there is a possibility that individuals in the middle or upper EI ability level are not correctly represented in the distribution and, thus the MSCEIT scores do not represent the actual distribution in EI ability. For a discussion of further methodological and theoretical questions regarding the MSCEIT, reference is made here to the authors Fiori and Vesely-Maillefer [12].

In addition, with regard to the ability EI, measured by the MSCEIT, the question arises as to how much it depends on personality traits [21]. Studies demonstrate that the MSCEIT is related with the Big Five factors but the correlations are low [61,62]. Brackett and Geher [63] indicate that the MSCEIT “is significantly related to, but not redundant with the BIG Five personality traits” (p. 35). In addition, Lopes et al. [64] found no correlation between the MSCEIT and public and private self-consciousness, mood, social desirability and self-esteem. A validation of the MSCEIT for the sports context has not yet been carried out.

Finally, as with regards to our finding on the relative effects of ability and trait EI on performance, the impact of trait EI on the self-assessment of athletic performance was independent of ability EI, while ability EI could not achieve any effects here either. Our findings are in line with the conclusions of Laborde et al. [37], who encourage using TEIQue in applied sports psychology because of its ability to predict neurophysiological outcomes and strong psychometric properties in athletics [28,37,51]. Due to our results, but also due to the practical handling and the good availability of the TEIQue, our study supports the idea that trait EI measured by the TEIQue is best suited for applied sport psychology.

## 5. Future Implications

Based on the current findings regarding our research questions, the following recommendations for future research have been identified. First, the operationalizing of sports performance across various types of sport continues to be a major challenge in sports psychology research. On the one hand, future EI investigations should look for comparable measurements of sports performance in different sports and at different levels of sports performance. On the other hand, we recommend that researchers should identify and operationalize emotionally relevant performance indicators or emotionally relevant dimensions of sports performance. Second, future research could attempt to categorize the type of sports in terms of emotional demands and carry out analyses on this basis. Third, for future research it would be desired to have studies that achieve sufficient sample sizes for different sports and all expertise levels in the respective sports in order to be able to carry out meaningful comparisons. Fourth, future studies related to EI and sports performance should use longitudinal and experimental designs to deeper establish the relationship. Fifth, future research should include a differentiated analysis of the subcategories of the EI for each sport or sports group. We believe that the relevance of individual branches of EI differs in each sport in contributions to the preceding outcome variables. Finally, our study indicates some evidence for the superiority of the trait EI in applied sports psychology. However, to confirm this result, replications of our study are necessary.

## 6. Limitations

In order to fully reflect our results, it is also necessary to consider some limitations of the current study. First, as this is a cross-sectional study, results do not provide evidence regarding causality. Second, challenges relating to our online survey were the sampling and response rates as well as non-respondent characteristics. The return rate (19.1%) was unfortunately low, although efforts were made to improve it. This could limit the generalizability of our study. Although a low response rate in online surveys does not automatically reflect a low level of representativeness [65], future research should find ways to improve the response rate of athletes. Third, as already mentioned, it is challenging to find common sports performance measures across sports and this may influence the impact of the results. However, we have used sports performance outcomes that seem appropriate and appear to be valid in sports [41,42]. Fourth, Cronbach’s alpha was below 0.70 for three out of four TEIque-SF subscales. This indicates a low internal consistency. However, in the present study, only the total score of the TEIque-SF was used. Sixth, the order of the instruments used may influence the responses and, thus, the results. For example, thought content may be activated during the processing of the first test, which may influence judgments on the second inventory. In addition, the subjectively perceived mental effort that may occur during the processing of the first test may have an influence on the response to the items of the subsequent questionnaire. Future studies should address this by randomizing the order of the instruments or examining a possible impact in more detail. Finally, we did not measure personality. Therefore, in our study we could not examine the incriminate validity with regard to personality traits (e.g., Big 5). Nevertheless, we were able to argue, with the help of relevant literature, why it can be assumed that overlapping of EI and personality traits is not to be expected in the sports context.

## 7. Conclusions

This is the first study with a direct comparison of ability and trait EI instruments regarding the prediction of sports performance. Results demonstrated that trait EI predicted the self-assessment of athletic performance. We have no evidence for the concept of the ability EI operationalized with the MSCEIT. Overall, it gives cause for optimism to consider trait EI as a potential predictor of self-assessment of athletic performance. It would seem that the trait model of EI is best suited for applied sport psychology, also due to the valid and easily handled instrument. Professionals working in practice, like trainers, athletes, sports managers and applied sports psychologists, need adequate expertise on the nature and the importance of trait EI for sports performance outcomes. This knowledge can help effectively shape the relationships with the various athletes individually and help them work optimally with each other to finally achieve peak performance. Finally, based on this expertise, practitioners and coaches can establish the implementation of EI screening and EI development programs as an integral part of the training process, professionalizing their work further. However, more differentiated explorations of EI with regard to the importance of each dimension for different types of sports are needed to identify the specific benefit for each sport. In addition, based on that, well-founded EI training programs for the emotional demands in different sports can then be developed accordingly.

## Figures and Tables

**Table 1 sports-09-00060-t001:** Descriptive statistics on socio-demographic information, performance levels and EI levels.

	Individual (*N* = 114)		Individual-O (*N* = 91)		Team (*N* = 118)		Global (*N* = 323)	
**Gender**	***N***	***%***	***N***	***%***	***N***	***%***	***N***	***%***
Female	52	45.6	36	39.6	64	54.2	152	47.1
Male	62	54.4	55	60.4	54	45.8	171	52.9
**Nationality**	***N***	***%***	***N***	***%***	***N***	***%***	***N***	***%***
AT	1	0.9	0	0	0	0	1	0.3
CH	1	0.9	0	0	0	0	1	0.3
GER	112	98.2	90	98.9	114	96.6	316	97.8
LU	0	0	1	1.1	4	3.4	5	1.5
**Marital status**	***N***	***%***	***N***	***%***	***N***	***%***	***N***	***%***
Single	61	53.5	54	59.3	68	57.6	183	56.7
Married	33	28.9	24	26.4	38	32.2	95	29.4
With a partner	20	17.5	13	14.3	12	10.2	45	13.9
**Highest completed level of education**	***N***	***%***	***N***	***%***	***N***	***%***	***N***	***%***
LSC	4	3.5	1	1.1	0	0	5	1.5
GCSE	7	6.1	13	14.3	6	5.1	26	8.0
VTD	6	5.3	7	7.7	5	4.2	18	5.6
GCE	57	50.0	34	37.4	48	40.7	139	43.0
BA/MA/D	31	27.2	25	27.5	55	46.6	111	34.4
Missing	9	7.9	11	12.1	4	3.4	24	7.4
**Sports performance**	***N***	***%***	***N***	***%***	***N***	***%***	***N***	***%***
*Level of expertise*								
National	68	59.6	63	69.2	87	73.7	218	67.5
International	4	3.5	9	9.9	20	16.9	33	10.2
Missing	42	36.8	19	20.9	11	9.3	72	22.3
*Sports success*								
National	89	78.1	80	87.5	89	75.4	258	79.9
International	25	21.9	11	12.1	29	24.6	64	20.1
*Self-assessment of athletic performance*	***M***	***SD***	***M***	***SD***	***M***	***SD***	***M***	***SD***
Range (4–20)	14.52	3.1	13.47	3.1	14.43	2.53	14.19	2.96
**EI measurement**	***M***	***SD***	***M***	***SD***	***M***	***SD***	***M***	***SD***
MSCEIT (*N* = 219)	98.40	15.9	97.91	17.79	101.76	15.10	99.85	15.97
TEIque-SF (*N* = 323)	5.26	0.60	5.14	0.59	5.22	0.56	5.21	0.58

Abbreviations: Individual: individual sports athletes without a direct opponent; Individual: individual sports athletes with a direct opponent; Team: team sports athletes; AT: Austria; CH: Switzerland; GER: Germany; LU: Luxembourg; LSC: Lowest school certificate; GCSE: General Certificate of Secondary Education; VTD: Vocational technical diploma; GCE: General Certificate of Education; BA/MA/D: Bachelor/Master/Diploma; *M*: Mean, *SD*: standard deviation; TEIQue: trait emotional intelligence questionnaire; MSCEIT: Mayer-Salovey-Caruso Emotional Intelligence Test. Note: Missing = no data reported respectively a clear allocation was not possible.

**Table 2 sports-09-00060-t002:** Logistic regression analysis results for the effects of ability and trait EI on sports performance.

	Predictor	*B*	*S.E.*	*χ* ^2^	*df*	*OR*	*p*	*CI 95%*	*Pseudo-R* ^2^
**Effects of Ability and Trait EI**	**Level of expertise in sports**								
	Intercept	−2.59	1.34	3.75	1	0.08	0.05		
	Ability EI	0.01	0.01	0.71	1	1.01	0.40	0.99–1.04	0.01
	Intercept	−2.23	1.64	1.85	1	0.11	0.17		
	Trait EI	0.07	0.31	0.05	1	1.07	0.83	0.58–1.97	0.00
	**Sports success**								
	Intercept	−1.04	1.03	1.01	1	0.36	0.32		
	Ability EI	−0.00	0.01	0.07	1	0.98	0.80	0.98–1.02	0.00
	Intercept	−3.62	1.31	7.63	1	0.03	0.01		
	Trait EI	0.43	0.25	3.00	1	1.53	0.08	0.95–2.48	0.02
**Relative effects of Ability and Trait EI**	**Level of expertise in sports**								
	Intercept	−3.55	2.30	2.37	1	0.03	0.12		
	Ability EI	0.01	0.01	0.59	1	1.01	0.44	0.99–1.04	0.06
	Trait EI	0.23	0.35	0.45	1	1.26	0.50	0.64–2.50	0.06
	**Sport Success**								
	Intercept	−2.64	1.87	2.00	1	0.07	0.16		
	Sports success Ability	−0.00	0.01	0.18	1	1.00	0.67	0.98–1.02	0.03
	Trait EI	0.19	0.29	0.43	1	1.21	0.51	0.69–2.12	0.03

Abbreviations: *B*: beta coefficient; *S.E.*: standard error; *χ2:* Wald; *df*: degrees of freedom; *OR*: Odds Ratio; *p*: *p*-value; *CI*: confidence interval; *R*^2^: Nagelkerke’s *R*^2^ (Pseudo-*R*^2^).

**Table 3 sports-09-00060-t003:** Linear regression analysis results for the effects of ability and trait EI on sports performance.

		Regression Coefficients					Regression Model				
	Predictor	*B*	*β*	*T*	*df*	*p*	*R* ^2^	*F*	*df^1^*	*df* ^2^	*p*
**Effects of Ability and Trait EI**	**Self-assessment of athletic performance**										
	Intercept	12.94		10.81	1	0.00					
	Ability EI	0.02	0.09	1.26	1	0.21	0.01	1.59	1	218	0.21
	Intercept	8.86		6.10	1	0.00					
	Trait EI	1.02	0.20	3.69	1	0.00	0.04	13.65	1	322	0.00
**Relative effects of Ability and Trait EI**	**Self-assessment of athletic performance**						0.04	20.16	4	218	0.07
	Intercept	8.79		4.25	4	0.00					
	Ability EI	0.01	0.08	1.23	4	0.22					
	Trait EI	0.80	0.17	2.49	4	0.01					

Abbreviations: *B*: beta coefficient; *β*: standardized beta coefficient; *T*: t- statistic; *df*: degrees of freedom; *p*: *p*-value; *R*^2^: coefficient of determination; *F*: F-statistics.

## Data Availability

The data presented in this study are available on request from the corresponding author. The data are not publicly available due to the data protection and policy standards.

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
