# Peer review of "Trait and Ability Emotional Intelligence and Its Impact on Sports Performance of Athletes"

_sports, 2021, doi:10.3390/sports9050060_

Round 1

Reviewer 1 Report

Dear author:

After check the data, please consider these comments in order to improve the text

LINE 1-43-149, in my opinion, this paragraph move to statistical part, NO IN PARTICIPANTS

LINE 156: 247 , TWO HUNDRED FORTY-SEVEN

LINE 159. SOCIDEMOGRAPYH DATA,,?. WHY THESE ITEMS..?. INCLUDE REFERENCE

LINE 212.  2.3 Procedure, Move to 2.2 Inmediatly after participants

LINE 245. Please explain if you check the normality of the data, to decide parametric or non parametric formulas, in case, which are variables..'

LINE 327, in the first paragraph (Discussion part), include the main results

LINE 347, "lead to small effect sizes", i don't undererstand this affirmation, please include reference or argument

LINE 448, 451, Please include in black colour

LINES 746, 468, These are ideas to complements on future, please include 1 paragraph with future lines

Line 505, include Ethical number and name of instittuion

Line 520, References, Include Doi:, IN ALL JOURNAL REFERENCES

In advanced

King Regards

Author Response

Response to Reviewer 1 Comments

We wish to thank you for the thoughtful and constructive comments on our manuscript "Trait and ability emotional intelligence and its impact on sports performance of athletes". We particularly appreciate your efforts to point us in the right direction for our revisions. The suggestions were very helpful when revising the manuscript. In this letter, we describe in detail how we addressed each of the issues you raised. We will reiterate all suggestions/comments/critiques in the order they were raised, followed by a description of our changes. To facilitate the review process, we use page numbers [in squared brackets] to indicate the exact part of the revised manuscript where we address each critique or suggestion for improvement.

Comments

Point 1: LINE 143-149, in my opinion, this paragraph move to statistical part, NO IN PARTICIPANTS.

Response 1. We appreciate this comment. We moved this paragraph to the section “Statistical data analysis” [p. 6-7; line 275-270].

Point 2. LINE 156: 247 , TWO HUNDRED FORTY-SEVEN.

Response 2. We thank you for this advice. We have corrected the mistake [p. 4; line 161].

Point 3: LINE 159. SOCIDEMOGRAPYH DATA,,?. WHY THESE ITEMS..?. INCLUDE REFERENCE

Response 3. We agree with the reviewer that this section leads to confusion. The sociodemographic data were collected as part of the study and are particularly relevant for replication studies. However, they will not be discussed again in the remainder of our manuscript, so we have removed this section. We also have removed sports performance information of the participants that is listed in this section, as this is explained specifically under “2.3.3. Sports performance” [p.5; line 208-230].

Point 4: LINE 212.  2.3 Procedure, Move to 2.2 Inmediatly after participants

Response 4. Thank you for this comment. We have moved the "Procedure" section and placed it behind the "Participant" section [p.4; line 163-176].

Point 5: LINE 245. Please explain if you check the normality of the data, to decide parametric or non parametric formulas, in case, which are variables..'

Response 5. We would like to thank you for this constructive comment. In the section "Statistical data analysis" we have added this point and described how we checked all assumptions for the linear multiple regression and why we had to use non-parametric tests [p. 6; line 270-279].

Point 6: LINE 327, in the first paragraph (Discussion part), include the main results

Response 6. Thank you for this comment. We have now added our main results into the first paragraph of the section “Discussion” [p. 9; line 265-270].

Point 7: LINE 347, "lead to small effect sizes", i don't undererstand this affirmation, please include reference or argument

Response 7. We agree with the reviewer that this paragraph leads to confusion. We have rephrased the sentence to make it understandable and we have added the corresponding reference [p.10; line 383-385].

Point 8: LINE 448, 451, Please include in black colour

Response 8. We thank you for this advice. The entire document was checked for black colour.

Point 9: LINES 746, 468, These are ideas to complements on future, please include 1 paragraph with future lines

Response 9. We have addressed this comment and have changed “Implications” to “Future implications” [p. 13; line 492].

Point 10: Line 505, include Ethical number and name of institution

Response 10. We have addressed this comment and added the number of Ethic approval. The name of the institution is listed [p. 13; line 554].

Point 11: Line 520, References, Include Doi:, IN ALL JOURNAL REFERENCES

Response 11. Thank you for this comment. We have added the digital object identifier (doi) to the references where an assigned doi was available [p.13-17].

We particularly appreciate your efforts to point us into the right direction for our revisions. The suggestions were very helpful when revising the manuscript. We hope that the revised manuscript has been improved in a way that warrants publication in Sports.

Reviewer 2 Report

General comments

The manuscript has the merit to provide useful information on the nature and the importance of EI to sports environment.

The manuscript is well-structured and logically conceptualized. The scientific soundness is high.

I would suggest the authors to address some minor comments aimed to improve the clarity of the text, making it more convincing.

Specific comments

Abstract

Line 13: “widely” comes first “differ” 

Introduction

 Line 67: please add a comma before “which”

Lines 74-77: do not keep this paragraph separate from the above one.

Line 120: move “to date” at the beginning of the sentence.

Lines 123-127: please rephrase for clarity.

M&M

Line 168: suggested rewording:”…only one used; it is currently…”

Discussion

Lines 350-354: use “should” instead of “must”. Then, please rephrase this part to improve the clarity of the text.

Lines 355-358: too long sentence. Perhaps, split it into two parts.

Line 376: “relates”

Line 378: please use the punctuation

Lines 408-411: This part is redundant. It might be appropriate deleting it.

Implications

Any practical advice to the practitioners and coaching staff? Please explore it.

Conclusions

Line 491: delete comma after EI.

Author Response

Response to Reviewer 2 Comments

We would like to thank you for the kind words and the recognition of our research topic, work and findings. Also we wish to thank you for the thoughtful and constructive comments on our manuscript "Trait and ability emotional intelligence and its impact on sports performance of athletes". We have addressed each point you have made. We particularly appreciate your efforts to point us in the right direction for our revisions. The suggestions were very helpful when revising the manuscript. In this letter, we describe in detail how we addressed each of the issues you raised. We will reiterate all suggestions/comments/critiques in the order they were raised, followed by a description of our changes. To facilitate the review process, we use page numbers [in squared brackets] to indicate the exact part of the revised manuscript where we address each critique or suggestion for improvement.

Comments

Point 1: Line 13: “widely” comes first “differ”

Response 1. We thank you for this advice. We have corrected the mistake [p. 1; line13].

Point 2. Line 67: please add a comma before “which”

Response 2. We thank you for this advice. We have added a comma before “which” [p. 2; line 67].

Point 3: Lines 74-77: do not keep this paragraph separate from the above one.

Response 3. We appreciate this comment. We have added this paragraph to the above paragraph [p.2; line 73].

Point 4: Line 120: move “to date” at the beginning of the sentence.

Response 4. We thank you for this advice. We have moved “to date” at the beginning of the sentence [p.3; line 121].

Point 5: Lines 123-127: please rephrase for clarity.

Response 5. We agree with the reviewer that this paragraph leads to confusion We have addressed this comment and have rephrased the sentence to make it understandable [p.3; line 126-133].

Point 6: Line 168: suggested rewording:”…only one used; it is currently…”

Response 6. We have addressed this comment and have changed the wording [p.4; line 187-188].

Point 7: Lines 350-354: use “should” instead of “must”. Then, please rephrase this part to improve the clarity of the text.

Response 7. We appreciate this comment. We have changed wording and we have rephrased the sentence to make it more understandable [p.10; line 389-392].

Point 8:  Lines 355-358: too long sentence. Perhaps, split it into two parts.

Response 8. We thank you for this advice. We have reduced the sentence and hopefully made it clearer [p.10; line 397-400].

Point 9:  Line 376: “relates”

Response 9. We thank you for this advice. We have corrected the mistake [p. 10; line 418].

Point 10:  Line 378: please use the punctuation

Response 10. Thank you for the note. We have added the punctuation [p.10; line 420].

Point 11:  Lines 408-411: This part is redundant. It might be appropriate deleting it.

Response 11. We thank you for this helpful comment. We have now reduced the paragraph [p.11; line 450-453].

Point 12:  Any practical advice to the practitioners and coaching staff? Please explore it.

Response 12. We would like to thank you for the advice to give more information on this issue. Now we added practical advice to the practitioners and coaching staff [p.13; line 538-543].

Point 13:  Line 491: delete comma after EI.

Response 13. We thank you for this advice. We have removed the comma after EI [p.13; line 533].

Again, we would like to thank you very much for sharing your thoughts and constructive comments. The detailed suggestions were very helpful. We hope that the revised manuscript has been improved in a way that warrants publication in Sports.

Reviewer 3 Report

Manuscript sports-1173264-peer-review-v2

This study investigated whether ability and trait emotional intelligence (EI) was associated with self-reported sport performance.  The authors used two validated instruments of ability EI and trait EI to survey a broad sample of athletes at a variety of competition levels and who participated in a variety of sports.  They found that trait EI positively predicted self-assessment of athletes’ performance whereas ability EI did not predict any outcome of sports performance. The effect of trait EI was independent of the ability EI.  The authors conclude that there is evidence for the superiority of trait EI in applied sports psychology.  The study appears to be well conducted, results are clearly presented, and the authors provide a thoughtful discuss and conclusions that are supported by the results.  The manuscript is generally well-written, although some moderate English editing throughout is needed.  Some additional considerations are given below.

Major Comments

  • Introduction is far too long. Although the authors provide a nice background and well-reasoned rationale for the study, this is far too detailed and extensive for a primary research article and could be reduced by half and still achieve an effective introduction.  Suggest substantial reduction, focusing on only the MOST relevant background, current knowledge gap, rationale, and aims of the study
  • Statement on ethical review/IRB review and informed consent appear to be missing
  • Analysis of whether the TYPE of sport influenced the relationship between EI and sport performance is an important consideration that appears to be missing. It is understood this appears to be a pilot trial, and the sample size is likely too small to be powered to detect any differences between sports, but this could be a source of variability and the authors should justify why it was not examined in the present study, even though they suggest this as a future direction in Section 5
  • Aside from continuous EI scores, were there clear cutoffs or percentiles that were associated with higher performance? (for example, were athletes in the 75th percentile or higher for trait EI higher performers? Or was the association stronger in higher or lower percentiles?)

Minor Comments

  • English language writing is generally good, but minor corrections to grammar, spelling, and English-language style are required throughout
  • Methods, Line 149 – 247 athletes participated regularly in competition – could you provide the % of the total sample that 247 represents?
  • Methods – inconsistent use of numerals (i.e., 323 in Line 143) and text numbers (i.e., two hundred forty-seven (Line 149); Fifty-six (line 150); please correct and standardize throughout
  • Methods, Lines 152-165 – why is this section italicized?
  • Methods, Line 163 – Order of tests not randomized, please explain why, as the order of tests may have influenced the findings
  • Methods, Line 180 and Lines 188 – in the former, Cronbach’s alpha is spelled out, in the latter, Cronbach’s α is written as a Greek letter – please revise for consistency
  • Methods, Lines 204-207 – how was “greatest success” defined, grouped, and analyzed? This seems like it could be a source of significant variability
  • Methods, Sections 2.3.1 and 2.3.1 – please provide copies of the complete MSCEIT and TEIQue-SF instruments as an online supplement and/or appendix
  • Methods, Line 248-249 is this a typo? Because of assumption violations, shouldn’t NON-parametric statistics should have been used?
  • Results, Table 1 – in Line 164 of the Methods, it states that 323 athletes completed the entire survey (demographics, MSCEIT, and TEIQue), but in Table 1 it shows that only 219 participants completed the MSCEIT and 323 completed the TEIque-SF. Why the discrepancy in sample size?
  • Results, Titles of sections 3.2, 3.3, and 3.4 – suggest changing “effects” (which implies causation) to “association”
  • Results – did the type of sport (i.e., skill, team, solo, endurance, strength, etc.) affect the associations between EI and sport performance? Or other relationships studied?
  • Discussion – would you expect different results if you administered the ability EI instrument immediately prior to an important competition (e.g., national championship, world championship, etc.) and analyzed whether it was associated with an athlete’s performance at that competition?

Author Response

Response to Reviewer 3 Comments

We would like to thank you for the kind words and the recognition of our research topic, work and findings. Also, we wish to thank you for the thoughtful and constructive comments on our manuscript "Trait and ability emotional intelligence and its impact on sports performance of athletes". We have addressed each point you have made. We particularly appreciate your efforts to point us in the right direction for our revisions. The suggestions were very helpful whilst revising the manuscript. In this letter, we describe in detail how we addressed each of the issues you raised. We will reiterate all suggestions/comments/critiques in the order they were raised, followed by a description of our changes. To facilitate the review process, we use page numbers [in squared brackets] to indicate the exact part of the revised manuscript where we address each critique or suggestion for improvement.

Comments

Point 1: Introduction is far too long. Although the authors provide a nice background and well-reasoned rationale for the study, this is far too detailed and extensive for a primary research article and could be reduced by half and still achieve an effective introduction. Suggest substantial reduction, focusing on only the MOST relevant background, current knowledge gap, rationale, and aims of the study.

Response 1. We appreciate this comment and agree with the reviewer that the introduction is far too long. We have made substantial reductions and tried to focus on the most relevant aspects [p.1-3].

Point 2. Statement on ethical review/IRB review and informed consent appear to be missing

Response 2. Thank you for the note. Statement on ethical review and informed consent were listed under the headings “Institutional Review Board Statement” and “Informed Consent Statement” [p.13-14; line 578, 581].

Point 3: Analysis of whether the TYPE of sport influenced the relationship between EI and sport performance is an important consideration that appears to be missing. It is understood this appears to be a pilot trial, and the sample size is likely too small to be powered to detect any differences between sports, but this could be a source of variability and the authors should justify why it was not examined in the present study, even though they suggest this as a future direction in Section 5.

Response 3. We fully agree with the reviewer that analyzing whether the type of sport influences the relationship between EI and sport performance is an important consideration. In our manuscript, we described that athletes interact in their “type of sport” and, therefore, athletes are influenced by the type of sport to which they belong. Furthermore, the properties of those groups are in turn influenced by the athletes who make up that group („Data preprocessing“ [p. 6; line 268-274]). There are several studies that have raised this question. The comparison of team sports and individual sports was especially focused on. However, according to the current state of the study, no differences have been found so far (Kajbafnezhad et al., 2011; Soflu et al., 2011; Laborde et al., 2014; Laborde et al., 2017). We were aware of the need to consider the influence of type of sports when investigating our research question. However, due to the sample size at group level, our sample was not suitable to perform multilevel analyses as actually desired, which would have allowed to analyze the influence of each type of sports on EI (e.g. in a total of 56 different type of sports, 23 were represented by only one athlete). For this reason, we classified type of sports into three groups, following the categorization of Laborde et al. (2018). The first group is “individual sports athletes without a direct opponent”. This includes sports like track or skiing. Laborde et al. (2018) assume that “here EI is particularly relevant to perceiving one’s own emotions, like fear or anxiety, and to regulating and using them where necessary to perform at one’s best” (p. 290). The second group is “individual sports athletes with a direct opponent” (e.g. boxer or tennis player). Here is the assumption that “in addition to the elements mentioned previously, when facing a direct opponent, it is advantageous to be able to perceive, regulate, and use the opponent’s emotions as well as one’s own emotions” (p.290). The final group is “team sports athletes” (e.g. footballers or basketballers) for which “it is particularly relevant for success to not only focus on one’s own emotions and the opponent’s emotions but also to perceive, regulate, and use the emotions of the teammates” (p.290). In all regression models in our study, we used dummy variables for these three groups to control the effect of the “type of sport” in our analyses. The study findings remained unchanged. We still believe that the "type of sport" is important for the analysis of the relationship between EI and athletic performance in competitive sports. However, we also believe that the categorizations made so far are not suitable to investigate this relationship. Rather, we suggest that future research should attempt to categorize sports in terms of emotional demands and conduct analyses based on this. We have pointed this out in our manuscript in the section “Future Implications“ [p.12; line 516-520].

Point 4: Aside from continuous EI scores, were there clear cutoffs or percentiles that were associated with higher performance? (for example, were athletes in the 75th percentile or higher for trait EI higher performers? Or was the association stronger in higher or lower percentiles?)

Response 4. Thanks for this comment. We assume that EI is a continuous construct and categorizing a continuous construct would only lead to a loss of information. We have therefore disregarded a categorization and analyzed all variables in their continuous distribution.

Point 5: English language writing is generally good, but minor corrections to grammar, spelling, and English-language style are required throughout

Response 5. We thank you for this note. We have had our manuscript checked once again with regard to grammar, spelling, and English-language style.

Point 6: Methods, Line 149 – 247 athletes participated regularly in competition – could you provide the % of the total sample that 247 represents?

Response 6. We thank you for this advice. We have added the % of the total sample that 247 represents [p.4; line 167].

Point 7: Methods – inconsistent use of numerals (i.e., 323 in Line 143) and text numbers (i.e., two hundred forty-seven (Line 149); Fifty-six (line 150); please correct and standardize throughout

Response 7. We have addressed this comment and we changed to consistent use of numerals [p.4; line 166-168].

Point 8:  Methods, Lines 152-165 – why is this section italicized

Response 8. We thank you for this advice. We have corrected the mistake [p.4; line 171-185].

Point 9:  Methods, Line 163 – Order of tests not randomized, please explain why, as the order of tests may have influenced the findings

Response 9. We agree with the reviewer that the order of tests is an aspect to be considered when conducting a survey with a broader test battery. We assume that the order of testing does not have a significant effect on the results. We are not aware of a single study showing this for EI. However, we have added this point to the discussion regarding the weaknesses of the study [p.13; line 562-567].

Point 10: Methods, Line 180 and Lines 188 – in the former, Cronbach’s alpha is spelled out, in the latter, Cronbach’s α is written as a Greek letter – please revise for consistency

Response 10. We thank you for this advice. We have corrected the mistake [p.5; line 205].

Point 11: Methods, Lines 204-207 – how was “greatest success” defined, grouped, and analyzed? This seems like it could be a source of significant variability

Response 11. We appreciate this comment. We have now added how sports success was grouped and analyzed [p.5; line 231-232].

Point 12:  Methods, Sections 2.3.1 and 2.3.1 – please provide copies of the complete MSCEIT and TEIQue-SF instruments as an online supplement and/or appendix

Response 12. We thank you for this advice. We will provide a copy of the complete TEIQue-SF instruments as an online supplement. The publishers of MSCEIT specifically point out that the test, including all of its parts, is copyrighted and any use without the consent of the publisher is illegal and liable to prosecution. For this reason, we are unfortunately unable to provide a copy of the MSCEIT.

Point 13:  Methods, Line 248-249 is this a typo? Because of assumption violations, shouldn’t NON-parametric statistics should have been used?

Response 13. This is absolutely right. This was a typographical mistake. Thank you for the extremely important advice. We have corrected it accordingly [p.6; line 290].

Point 14:  Results, Table 1 – in Line 164 of the Methods, it states that 323 athletes completed the entire survey (demographics, MSCEIT, and TEIQue), but in Table 1 it shows that only 219 participants completed the MSCEIT and 323 completed the TEIque-SF. Why the discrepancy in sample size?

Response 14. We are thankful for the opportunity to explain this point in more detail. While the procurement as well as the evaluation of the TEIque is simple and free of charge, the MSCEIT is a chargeable instrument. For the test execution, the test manual, a software (807 Euro) and the individual evaluations must be purchased. Accordingly, costs are incurred for each individual evaluation (e.g. 740 Euro for 50 evaluations). We had ordered a contingent of 400 evaluations in addition to the software. There were 2 years between the procurement and our online survey. In this time an update of the MSCEIT software was carried out, which required a new procurement. the supplier could not offer us any other possibility. These additional costs of almost 7000 Euros were not included in our budget, so we had to decide for a reduced number of evaluations (220). Therefore, no evaluations of the MSCEIT could be performed for 103 participants. In addition, one evaluation resulted in an error, thus, it could not be included in our study. The selection of the 219 participants was random.

Point 15:  Results, Titles of sections 3.2, 3.3, and 3.4 – suggest changing “effects” (which implies causation) to “association”

Response 15. We have addressed this comment and have changed “effects” to “association” [Titles of sections 3.2, 3.3, and 3.4; p. 8; line 329; p. 9; line 349,356].

Point 16: Results – did the type of sport (i.e., skill, team, solo, endurance, strength, etc.) affect the associations between EI and sport performance? Or other relationships studied?

Response 16. That is a very important comment. Thank you very much for mentioning this. In our study, we tried to take this point into account by coding the type of sports into three groups: individual sports athletes without a direct opponent, individual sports athletes with a direct opponent and team sports athletes, following the categorization of Laborde (2018). In all our analyses, we controlled whether the associations changed as a function of the different groups of sports. However, this was not the case, the results remained unchanged. Therefore, for our study we can conclude that type of sports according to our classification (individual sports athletes without a direct opponent, individual sports athletes with a direct opponent and team sports athletes) does not affect the associations between EI and sport performance. We have added this information in the results section [p. 10; line 373-375]. Other relationships were not investigated within the scope of our study.

Point 17:  Discussion – would you expect different results if you administered the ability EI instrument immediately prior to an important competition (e.g., national championship, world championship, etc.) and analyzed whether it was associated with an athlete’s performance at that competition?

Response 17. Thank you for this interesting question. While it can be considered certain that trait EI, conceived as a stable trait that does not change quickly depending on the context, is more likely to have a stronger influence on long-term outcomes in comparison to short-term outcomes (Laborde, 2016), we are not aware of any studies that specifically address this question with regard to ability EI (Laborde, 2016). Therefore, we can only speculate on this quite interesting question.

A little thought experiment here. As we have listed in the discussion section, we assume that the ability EI assessed by MSCEIT basically measures the knowledge dimension of EI, and there can be discrepancies between that and the actual day-to-day application of this knowledge in real social-emotional training and competition interaction. Perhaps it could be conceivable that by answering the MSCEIT immediately before an important competition, athlete's existing knowledge about competition-relevant emotional strategies, is activated and receives attention, which may have a positive effect on his or her personal athletic performance in that competition. However, this needs to be investigated in appropriate experimental studies.

Again, we would like to thank you very much for sharing your thoughts and constructive comments. The detailed suggestions were very helpful. We hope that the revised manuscript has been improved in a way that warrants publication in Sports.
